# Extracellular Vesicles from Adipose Tissue Could Promote Metabolic Adaptation through PI3K/Akt/mTOR

**DOI:** 10.3390/cells11111831

**Published:** 2022-06-03

**Authors:** Jaime Delgadillo-Velázquez, Herminia Mendivil-Alvarado, Carlos Daniel Coronado-Alvarado, Humberto Astiazaran-Garcia

**Affiliations:** 1Coordination of Nutrition, Research Center for Food and Development (CIAD), Ave. Gustavo E. Astiazarán #46, Hermosillo 83304, Mexico; jaime.delgadillo.220@estudiantes.ciad.mx (J.D.-V.); herminia.mendivil@estudiantes.ciad.mx (H.M.-A.); coronado222@estudiantes.ciad.mx (C.D.C.-A.); 2Departamento de Ciencias Químico Biológicas, Universidad de Sonora, Hermosillo 83000, Mexico

**Keywords:** cell communication, exosomes, obesity, insulin resistance

## Abstract

Extracellular vesicles (EVs) are nanoparticles secreted by cells under physiological and pathological conditions, such as metabolic diseases. In this context, EVs are considered potential key mediators in the physiopathology of obesity. It has been reported that EVs derived from adipose tissue (ADEVs) contribute to the development of a local inflammatory response that leads to adipose tissue dysfunction. In addition, it has been proposed that EVs are associated with the onset and progression of several obesity-related metabolic diseases such as insulin resistance. In particular, characterizing the molecular fingerprint of obesity-related ADEVs can provide a bigger picture that better reflects metabolic adaptation though PI3K/Akt/mTOR. Hence, in this review we describe the possible crosstalk communication of ADEVs with metabolically active organs and the intracellular response in the insulin signaling pathway.

## 1. Introduction

Extracellular vesicles (EVs) are nanoparticles released by cells which are enclosed by a lipid bilayer. It is well known that EVs play a role in intercellular communication through proteins, lipids, and genetic material; they also serve as vehicles to release degradable content [1,2]. Traditionally, EVs are classified according to their biogenesis into two subgroups, exosomes and ectosomes; the former are derived from endosomal pathway and have a size range of 40 to 160 nm. In contrast, the latter are derived from the plasma membrane and typically measure 50–1000 nm [1,2,3]; depending on their size, they can also be called small or large EVs, respectively [4].

Not much is known about specific markers that make it possible to distinguish the subtypes of EVs. Additionally, their heterogeneous composition and varied size has not allowed the precise identification of the subgroup to which they belong [2,5]. Another limitation to identify EVs subtypes, is that their size and density is similar to that of lipoproteins (HDL, LDL, VLDL) [6]. However, once isolated and purified from the extracellular medium or from biological fluids using methods such as ultracentrifugation, precipitation or even molecular exclusion chromatography, their size can be determined. Finally, EVs’ specific proteins linked to their origin can be analyzed [1,7,8]. 

## 2. Biogenesis and Uptake of EVs

EVs originate from multiple mechanisms. For exosomes, the endosomal sorting complex required for transport (ESCRT) is required, including Alix and Tsg 101 proteins, which are necessary to form intraluminal vesicles by grouping them within multi-vesicular bodies [3]. Then, these are eventually degraded by lysosomes or fused with the plasma membrane (PM) and secreted as exosomes [1,3]. Other known alternative pathways to ESCRT include transmembrane proteins called tetraspanins (CD9, CD63 and CD81) and the enzyme sphingomyelin phosphodiesterase 3 [2]. These proteins play an important role in the invagination of PM which is necessary in the formation and secretion of exosomes [2,9,10]. In contrast, microvesicles require changes in the redistribution of PM lipids in addition to the ESCRT machinery [6]. The changes are induced by flippases and translocases, enzymes that are activated by the increase in the cytosolic concentration of Ca^2+^ [11]. As a result, phosphatidyl serine is repositioned to the outer side of the PM. At the same time, the increase in calcium activates Rho-II kinase proteins (responsible for phosphorylating myosin chains) [12], followed by binding with actin. These proteins, combined with enzymes such as syntenin-1 and syndecane-1, cause reorganization and contraction of the cytoskeleton, protrusion of PM, and finally detachment of these vesicles into the extracellular medium [13].

Once EVs have been released, they can reach neighboring or distal cells by bordering the cell surface until they reach the membrane receptor of the target cell [1,4]. The biological signal is transmitted upon EVs binding to specific ligands, activating signaling cascades or merging directly with the PM to enter the cytoplasm [11]. Another way to uptake EVs is through endocytosis or phagocytosis, with a lipid membrane or raft receptor, generating multivesicular bodies that eventually release their functional content to the cytoplasm or are transported to the nucleus [14,15,16]. EVs transport different biomolecules: proteins (membrane, cytosolic and enzymes), lipids (phospholipids, diacylglycerol, ceramides, fatty acids), and genetic material (DNA, mRNA, miRNA, LncRNA), which can exert phenotypic changes in the target cell [1,6,17]. 

## 3. Functional Role of EVs

In mammals, EVs can take part in a variety of fundamental cellular processes such as immunoregulation, in which dendritic cells secrete exosomes that activate the T cell-mediated immune response [18]. Additionally, it has been reported that vesicles originated from Treg cells suppress inflammatory response of Th1 cells [19]. The EVs together with coagulation factors promote thrombin generation, induce cell proliferation, and increase capillarity in endothelium cells; thus, EVs are involved in hemostasis [20,21]. Finally, Oggero et al. [22] showed the role of EVs in tissue repair processes and as vascular inflammation mediators.

The literature suggests that EVs participate in cell-to-cell communication, see review [22], thus suggesting their role as agents of metabolic regulation. In particular, regulation of energy homeostasis takes place mainly in the brain, pancreas, liver, skeletal muscle, and adipose tissue [23]. Obesity is dysregulation of energy metabolism, leading to changes in the composition of tissue microenvironment and causing tissue dysfunction [24], affecting the biosynthesis, secretion, and signaling of soluble and insoluble molecules related to diseases such as insulin resistance [25,26]. 

## 4. Adipose Tissue

Adipose tissue (AT) is a specialized connective tissue that is classified into three types: white (WAT), beige, and brown (BAT) adipose tissue [27]. WAT is responsible for storing excess fat in the form of triacylglycerides, while BAT transforms chemical energy into heat form by high levels of decoupling protein expression 1 (UCP-1), regulating body temperature, and oxidizing fatty acids. Beige adipose tissue metabolizes more lipids than WAT, but has lower levels of UCP-1 expression [27,28]. 

Based on the location, the WAT can be subdivided into visceral and subcutaneous fat. Subcutaneous WAT is associated as a protective factor for the development of obesity and metabolic diseases [29,30]. Here, to better understand the pathological processes of obesity, we will focus on the visceral subtype, for which there is a large body of evidence of metabolic dysregulation [23,25,27]; however, it is not well understood how obesity is the main cause of AT dysfunction.

In physiological conditions, two of the three parts of AT are mature adipocytes, and the remainder is the stromal vascular fraction (SVF), which is composed of extracellular matrix and immune system, mesenchymal cells, fibroblasts, preadipocytes, endothelial cells, smooth muscle, and adipose tissue resident macrophages (ATM) [27]. ATM comprise 5 to 10% of SVF and can reach up to 50% in severe obesity [31]. Additionally, they secrete anti-inflammatory cytokines such as IL-10, IL-4, damper lipids, and remodel membrane lipids, along with the extracellular matrix [32]. 

Hence, Schenk et al. [33] assigned the pro-inflammatory profile to the M1 recruited macrophages and CD8+ T cells in the AT, because these cell types secrete cytokines that affect the cascade of insulin signaling. It is likely that the increase in lipolysis and secretion of free fatty acids due to the activation of hormone-sensitive lipase (HSL) by the AT, creates a vicious cycle that persists and further aggravates the catabolic state leading to metabolic syndrome or type 2 Diabetes (T2D). Therefore, ATM promote the proper functioning of lipid metabolism and glucose homeostasis [33]. In a fasting state, free fatty acids are used by the skeletal muscle as an energy substrate, and the glycerol produced is metabolized by the liver to provide glucose to the brain, a process called lipolysis. Under insulin-stimulated conditions, such as the postprandial state, ATM indirectly promotes re-esterification of circulating lipids to AT, a process known as lipogenesis [34,35].

Regarding adipocytes, they are cells that range from 50 to 100 μm in diameter; those in WAT have a single lipid droplet and fewer mitochondria than BAT, which allows them to store energy more efficiently than oxidize it [30]. Mature adipocyte regulation processes such as hyperplasia and hypertrophy lead to an increase in the number and size of adipocytes, respectively [32]. To perform the above functions, preadipocytes must be differentiated into mature adipocytes by a process called adipogenesis [36]. Several well-described transcription factors and molecules are involved in this process such as C/EBPs and PPARγ [37]. Interestingly, using a murine model, Zhang et al. [36] and Son et al. [37] showed that exosomes from AT promote and modulate adipogenesis in adipose mesenchymal cells, resulting in a coordinated process inside the microenvironment of cells conforming AT. 

During obesity, the maximum expansion capacity of the adipocyte is reached, when ATM produces pro-inflammatory cytokines that affect insulin signaling, causing lipotoxicity and leading to ectopic fat accumulation in tissues like liver and skeletal muscle [35,38]. Immuno-metabolic alterations of AT contribute to metabolic dysfunction and exacerbate the state of inflammation, negatively impacting homeostasis [29,39]; further research is needed to elucidate the molecular mechanisms of local and systemic insulin resistance (IR). Evidence suggests that EVs derived from AT are more a cause of metabolic disorder than a consequence, given that their molecular content can modulate signaling pathways and gene expression in target cells/tissues [34,40] or indirectly through micro RNAs from AT [41,42]. For this reason, EVs derived from AT (ADEVs) would be considered endocrine mediators of metabolic homeostasis.

## 5. ADEVs

Dietze et al. [43] studied in vitro adipocyte communication from breast tissue explants of women with BMI 22–29 Kg/m^2^ alongside myocytes from normal weight individuals, and they found that when pro-inflammatory cytokines and resistin were placed on the medium, no difference in skeletal muscle IR development was observed. This suggested that the onset and progression of obesity related IR is not only influenced by hormones but by other adipocyte-typical factors released into the medium. This opened a new research area to identify the “other factors”, which we now know as adipokines.

Later, Aoki et al. [44] analyzed EVs secreted by adipocytes in vitro. They found that larger adipocytes supplemented with high concentrations of insulin and glucose in the medium, secreted both larger size and amounts of EVs, which the authors called adiposomes. Those conditions simulated an obesity environment affecting the morphology of EVs. However, other factors that also interfered were the positive and negative controls used in the experiment (N-acetyl cysteine and TNF-α, respectively), showing that various intracellular conditions can trigger a functional response mediated by EVs [44]. 

In addition to the role of pro-inflammatory cytokines, it is postulated that IR is initiated at the AT due to obesity and chronic low-grade inflammation extends to other tissues [40,45]. The underlaying mechanisms have not been entirely elucidated, but evidence suggests that it is the genetic material contained in the ADEVs that regulates the phenotype of neighboring cells. Since ADEVs are present in circulation, they could regulate signaling and metabolic pathways of different cells/tissues [46]. This is the case of microRNAs (miRNAs) which can modulate the expression of specific genes, resulting in decreased protein translation [47,48]. In this regard, Muller et al. [49] found in primary adipocytes of a murine model, a profile of miRNAs contained in microvesicles differ in the group of hypertrophied adipocytes compared to the group of adipocytes in normal state. Authors suggested that obesity could affect the secretion and type of miRNAs [49].

Mori et.al. [50] inactivated the protein ADipo, which is responsible for the biogenesis of miRNAs in AT, and they found that the basic functions of adipocytes were altered. This resulted in WAT dystrophy that triggered ectopic accumulation of fat and BAT whitening. In addition, this in vitro inactivation confirmed that brown preadipocytes have characteristics of the WAT phenotype developing the severe IR model that lead to metabolic alterations [50]. 

Interestingly, one study showed that the profile of miRNAs of circulating exosomes changed after undergoing bariatric surgery [51]. Bioinformatics analysis of these molecules suggests insulin-related signaling pathways, finding higher HOMA-IR index value through biochemical analyses before surgery and a decrease after it. Thus, proving that exosomes derived from visceral AT are involved with glucose homeostasis. Finally, the authors analyzed the fatty acid binding protein 4 (FABP4) and it was found to be enriched, suggesting FABP4 as a possible biomarker of exosomes derived from AT [51].

### 5.1. Microenvironment in Adipose Tissue

Genomic analyses on miRNAs profiles present in serum, plasma, or explants are suggested as markers of molecular mechanisms involved in obesity, alongside with the expression of miRNAs contained in EVs [40,42,51]. First, we will focus from a paracrine regulation perspective and then endocrine fashion, that is to cells/tissues distal to AT as a model of cellular inter-communication. 

As previously mentioned, the IR development of AT depends on immuno-metabolic regulation; in this respect, communication between cells, specifically between adipocytes and ATM, is important, especially in obesity [41]. When the ATM population reaches 50% of SVF [31], most ATM change from an anti-inflammatory M2 phenotype to a pro-inflammatory M1 phenotype, leading to high concentrations of cytokines and EVs containing miRNAs being secreted, which in the end can trigger IR [38,40]. Hence, in a murine model of obesity where the animals were fed a high-fat diet, Zhang et al. [52] found a higher expression of miRNA-155 in microvesicles derived from the adipocyte and ATM; additionally, using in-silico analysis authors found SOCS1 to be one of miRNA155 target, thus activating STAT1 while simultaneously inhibiting STAT6. Furthermore, the effects of such a change on miRNA-155 silencing were restored. In addition, microvesicles derived from bone marrow macrophages over expressing miRNA-155, negatively impacted the insulin signaling and glucose uptake of adipocytes. These results demonstrated the intercommunication of both AT cells and that, in a state of obesity, adipocytes both affected and were affected by the same content of miRNA-155 in microvesicles [52].

By simulating the intercellular environment with respect to the function of AT with obesity, Yao et al. [53] using a co-culture of adipocytes and macrophages found a different mechanism by which AT with obesity promoted the M1 phenotype through miRNA-27a, analyzing the proteins and pro-inflammatory cytokines involved. Similarly, using a murine model, they overexpressed miRNA-27a, and the results showed that glucose uptake was affected and IR was promoted in a time-dependent way by exposure to the high-fat diet in obese mice; the results were reversed when miRNA-27a was inactive. Results suggest a new inhibition mechanism in the expression of PPARγ as a possible target gene regulated by miRNA-27a contained in exosomes and the subsequent activation of NF-kB [53]. 

Recently, Pan et al. [54] observed that the expression of miRNA-34a in epididimal WAT of a murine model with obesity was higher than the control group after 6 weeks of dietary intervention. Additionally, when analyzing SVF they observed that adipocytes express miRNA-34a. By simulating different in situ obesity conditions, the results indicated a dose-time-dependent response initiated from TNFα regulation and palmitic acid. Gene Klf4 is affected by miRNA-34a; inhibition of miRNA-34a promotes the M2 phenotype of ATM, while its over expression increases the expression of pro-inflammatory proteins characteristic of phenotype M1 associated with IR. This suggests that nutrient stress in hypertrophied adipocytes involves a paracrine response to ATM through exosomes, developing inflammation of AT due to miRNA, and thus systemically affecting glucose homeostasis [54].

### 5.2. Distal Intercom from Adipose Tissue

Here we explore the role of miRNAs contained in ADEVs in IR, through endocrine-like communication with other cells. In particular, Thomou et al. [42] showed that inactivation of ADipo, decreased the expression of circulating miRNAs; additionally, by administering BAT exosomes to an in vivo model restored the metabolic alterations associated with IR (vs. control group). BAT exosomes contain miRNA-99b, which directly affects the expression of the protein growth factor of fibroblasts 21 (FGF21), a key protein in glucose homeostasis of distal organs. The authors concluded that the exosomes derived from AT act as new adipokines [42]. 

Using primary hepatocytes from a diet-induced obesity model, Ying et al. [55] observed that ATM secretes exosomes related to glucose homeostasis. Additionally, they found that the expression of miRNA-155 induced IR. Later by either inactivating miRNA-155 expression in a murine model or by administering exosomes derived from the control group to the obesity group, authors showed that insulin sensitivity was reestablished using both methodological approaches, thus validating miRNA-155′s role in IR. They also found that miRNA-155 negatively regulated the expression of PPARγ and this, in turn, the protein GLUT4, implying that the effects had an impact on the cascade of insulin signaling in adipocytes, myotubes and hepatocytes [55].

Furthermore, Castaño et al. [56] found that a different group of miRNAs also affected the regulation of PPAR isoforms increasing hepatic triglycerides and glucose intolerance [56]. Differences between both studies [55,56] are likely due to different methodological approaches and the exosomes’ origin, as well as different posology in their administration.

Dang et al. [57] also evaluated AT exosomes and their role in hepatocytes from different murine models of obesity. AT exosomes were obtained from (a) ob/ob genetic model and (b) high-fat diet. Through massive sequencing, they found and validated that expression of miRNA-141-3p was lower in obesity groups compared to the control group. In models of obesity and regardless of their origin (genetic or diet-induced), exosomes were captured by in vitro hepatocytes and miRNA expression was accompanied by increased phosphorylation of the Akt protein [57], a serine/threonine-specific protein kinase that is key in transduction signaling pathways, especially those associated to insulin [29,35]. The results in [57] were validated with an exosome biogenesis inhibitor, leading to lower insulin sensitivity and therefore lower glucose capture. Along with bioinformatic analyses, the authors demonstrated that miRNA-141-3p affected PTEN protein, a phosphatase that dephosphorylates PI3K, involved in the insulin signaling cascade [57].

Just as there is communication with the liver, miRNAs contained in ADEVs can modulate gene expression by altering the phenotype in skeletal muscle. In this regard, Wang et al. [58] evaluated adipose-derived miRNAs and found higher concentration of miRNA-130b in a diet-induced obesity mice group, as well as in serum of people with obesity vs. a control group, respectively. They validated that the expression of this miRNA was increased during adipogenesis, by bioinformatic and in vitro analyses of 3T3-L1 cells. They showed that the target gene of miRNA130-b was the coactivator 1α of PPARγ (PGC1-α), involved in fatty acid oxidation and mitochondrial biogenesis. In addition, TGF-β activated the secretion of miRNA-130b in adipocytes; this was further validated using a C2C12 myotubes culture. The results suggest that AT may regulate the IR phenotype in the skeletal muscle [58] and may be through miRNA contained in ADEVs.

As discussed above, ATM derived from exosomes can regulate IR through miRNA-155 to PPARγ, which is involved in insulin signaling cascade proteins in skeletal muscle cells [55]. However, Yu et al. [59] found that it was miRNA-27a that regulated the expression of PPARγ, an miRNA with higher serum concentration in subjects with obesity; additionally, the authors showed a positive correlation between serum miRNA-27a concentration and BMI, and fasting glucose, respectively. Using different in vivo high fat diet and in vitro models of C2C12 cells treated with palmitate, inactivation of miRNA-27a improved glucose and insulin tolerance. Here, they demonstrated that the expression of exosomes containing miRNA-27a was derived from adipocytes rather than macrophages, and that in turn, the exosomes were captured by C2C12, concluding that in obesity, it is through exosomes derived from adipocytes that IR is developed in the skeletal muscle [59]. 

Similarly, a study by Gao et al. [60] showed that adipocytes secrete exosomes containing MALAT1, and evaluated the mTOR signaling pathway in POMC neurons. In the in vitro model, they were able to demonstrate that exosomes derived from adipocytes from mice with diet-induced obesity, alter the aforementioned pathway, resulting in weight gain associated with hyperphagy [60]. Alternatively, by transferring exosomes from the control group to the obese the group, results showed attenuated weight-gain and decreased appetite. In addition, in an in vitro co-culture model between adipocytes and neurons they were able to confirm that these neurons captured adipocyte exosomes simulating physiological conditions of intercellular communication. These results showed that through miRNA-181b and miRNA-144, mTOR, a signaling pathway involved as a nutrient-sensing protein, can regulate the development of obesity, and is modulated by these miRNAs. However, proteins and clinical indices associated with lipid metabolism and glucose homeostasis were not evaluated [60]. 

It has also been reported that ADEVs can be involved in the development of cardiovascular diseases [61,62]. However, more studies with the same methodology and techniques are needed to analyze multiple miRNAs that corroborate in a different population, the effects and mechanisms associated with IR in obesity through ADEVs, while also assessing protein and/or lipid content. In this respect, Kranendonk et al. evaluated exosomes from human explants of (a) subcutaneous and (b) visceral AT that were ectopically administered to the culture medium of hepatocytes and myocytes. To associate the IR response, they evaluated a profile of adipokines contained in the ADEVs of both groups and these were the following: IL-6, MCP-1, MIF, RBP-4 and resistin. By linking the adipokine concentrations to the phosphorylated Akt value, they found that MCP-1 was better correlated with subcutaneous and MIF and IL-6 than with the visceral region, only in the hepatocyte group. These data suggest that ADEVs may secrete pro-inflammatory cytokines that negatively affect the phosphorylation of Akt in the liver for the development of IR. In addition, the authors identified the over-expression of adiponectin in both explants, suggesting it as a possible marker of ADEVs [63].

In relation to the role of ATM in the development of IR, Song et al. [64] analyzed in vitro 3T3-L1 adipocytes stimulated with high concentrations of insulin and glucose. They assessed proteins contained in exosomes derived from adipocytes and identified that the Sonic Hedgehog (Shh) protein concentration was higher than Insulin resistant-free adipocytes. In addition, they noted that in stimulated adipocytes, IRS-1 serine phosphorylation was higher and time-dependent, and Akt phosphorylation and GLUT4 translocation were decreased. The authors also evaluated the role of in vitro IR adipocyte-derived exosomes in macrophages and identified that the response was polarized to the M1 phenotype, and that this phenotype was activated by the Ptch/PI3K signaling pathway in macrophages; effects were validated following the inactivation and inhibition of Shh. Finally, macrophages grown with exosomes from the IR group were added in co-culture with mature adipocytes and they observed that the expression of the IRS-1 and HSL proteins was reduced. These results imply a new mechanism by which adipocytes within an obesity environment communicate with ATM and induce the M1 phenotype and, in turn, they respond to neighboring adipocytes affecting insulin signaling through Shh, a protein contained in exosomes [64]. Likewise, evidence from in vitro models suggests that adipocytes stimulated in conditions simulating obesity and IR, communicate with adjacent adipocytes promoting hypertrophy and affecting insulin signaling through EVs [65,66].

Regarding other AT cells, Zhao et al. [67] conducted a study using a murine model where they analyzed how exosomes derived from ADSCs from the normal weight control group and administered to the diet-induced obesity group. Here, weight gain during the last stage of dietary intervention and the percentage of body fat decreased in the in vivo model vs. the obese control group. Furthermore, when they analyzed the tissues at the end of the intervention stage, using tissue from epididymal and inguinal fat deposits, they found a higher number of beige adipocytes and a smaller WAT size vs. the obese control group. Similarly, they identified that these effects were exerted on the macrophages by modulation of phosphorylation of STAT3 leading to the activation of Arg-1; by inactivating STAT3 the expression of Arg-1 decreased significantly, thus validating the proposed mechanism. Additionally, Zhao et al. [67] isolated macrophages from WAT in vitro onto a culture medium, and they noted that phenotype M2 was promoted by an increase in the expression of Arg-1 and IL-10 and the pro-inflammatory response of iNOS, TNF-α, and IL-12 was mitigated. These led the authors to infer that there was a mechanism of intercommunication of AT cells, which responded in a coordinated way to the regulation of obesity and promote systemic sensitivity to glucose and insulin [67]; furthermore, by injecting exosomes derived from ADSCs onto an obese mice group, they provided groundbreaking evidence for future treatment of obesity and IR.

In another study, Eguchi et al. [68] obtained EVs from the plasma of an obese mouse model and treated a culture of adipocytes with these EVs promoting the activation of macrophages M1, which resulted in higher concentration of pro-inflammatory cytokines. They also found a higher plasma concentration of Perilipin A in EVs from the obese vs. the control group, respectively; additionally, Perilipin A’s concentration was higher in exosomes potentially from ADEVs of obese vs. normal-weight subjects, suggesting this protein as a potential biomarker for these EVs [68]. In this regard, Connolly et al. [69] evaluated proteins contained in pre- and post-adipogenesis adipocytes using an in vitro model, they found a higher secretion of EVs within the size range of exosomes, in the pre-adipogenesis group. In addition, FABP4 and PREF-1 were more abundant in the pre-adipogenesis group. On the other hand, they observed higher levels of adiponectin in the post- vs. pre-adipogenesis group. These results can be used to work with adiponectin and FABP-4 as markers of exosomes derived from the adipocyte [69]. 

In another report by Durcin et al. [70], the content of proteins in EVs from adipocytes was analyzed, and they were classified by size into small (sEV) and large (lEV), with an average diameter of 100 nm and 100–180 nm for sEV and lEV, respectively. The results showed a larger production of sEV per adipocyte compared to lEV, which means that exosomes are preferably released compared to larger microvesicles. Using a proteomic-bioinformatic analysis of these EVs, different subcellular locations, biological processes, and molecular functions of proteins were identified for sEV and lEV. They also validated markers such as Alix, Tsg101, tetraspanins, CD9, CD63, and CD81 for exosomes; in the case of microvesicles, they assessed Flotyline-2 and Caveoline-1. The evidence of these markers in EVs will allow researchers to characterize them sEV and lEV [70]. Similarly, another study analyzing the mechanism with a genetically modified murine model of communication between endothelium cells present in SVF of AT and adipocytes, found Caveoline-1 in exosomes derived from endothelium cells. Dependent on the physiological state, Caveoline-1 increased in the fasting state vs. fed, suggesting a mechanism in which glucagon participates. Influencing the transfer of EVs in plasma. These results highlight the role of EVs in fasting/postprandial state, suggesting another marker involved in ADEVs and opening up an area for future research [71].

Proteomic analysis is used to characterize cells or tissues; however, the interpretation of coupled functional analysis and signaling pathways on metabolic and cellular processes derived from ADEVs, is more challenging [72] as shown in Figure 1. Lee et al. [73] determined the proteomic footprint of EVs derived from primary adipocytes focusing on the molecular footprint of T2D using two animal models: Otsuka Long-Evans Tokushima fatty (OLETF) and Long Evans Tokushima Otsuka (LETO) rat models of type 2 diabetes and obesity, respectively. Results showed higher concentrations of caveoline-1, aquaporin 7, and lipoprotein lipase in OLETF vs. LETO, suggesting high intracellular lipid traffic that could be used as a T2D biomarker [73]. One of the limitations identified in this study is the lack of a normal weight control group, thus limiting the ability to conclude on potential EVs’ mechanisms on the development of IR; however, their global approach can be used in diet-induced obesity models.

Interestingly, Hartwig et al. [74] characterized the secretome of EVs from primary culture of subcutaneous AT of women with normal weight or overweight; EVs were derived from adipocytes and compared to protein databases reported in ExoCarta and subsequent bioinformatic analysis. They found more than 800 exoadipokines involved in metabolic and signaling processes; however, from existing databases authors were not able to compare their results due to lack of specific markers of EVs from adipocytes or AT in humans. Additionally, 67 exoadipokines had not been previously identified, and of these, the characteristic was that they did not have a signal peptide, in other words, they were found to be exclusive to EVs [74]. When analyzing a target tissue’s proteome within EVs, it is complicated to differentiate whether those markers are produced in situ or if they were up-taken from EVs from a distal tissue. In addition, given that the proteome is a dynamic entity and specific to metabolic conditions, it is complicated to characterize and couple with different pathways of intercellular communication to elucidate pathological mechanisms, specifically in obesity. 

**Figure 1 cells-11-01831-f001:**
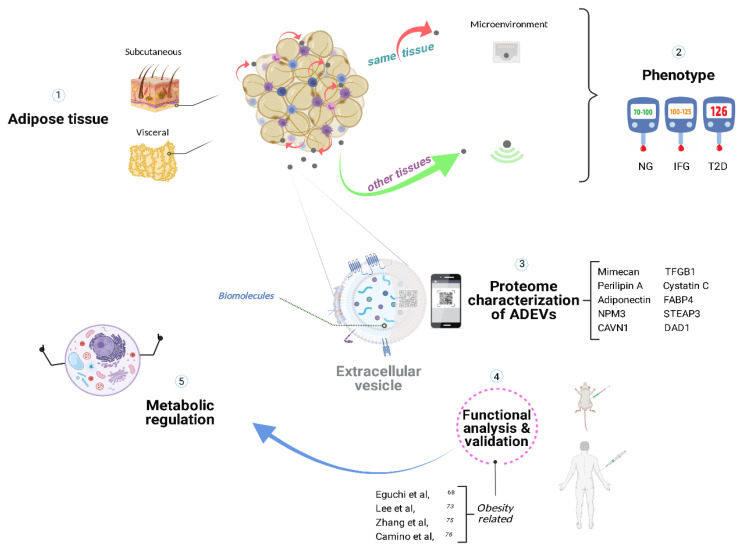
Metabolic homeostasis with the assistance of adipose-derived extracellular vesicles. Adipose tissue secretes extracellular vesicles that could participate in pathophysiological processes specific to the microenvironment in a paracrine manner and communicate with other tissues in an endocrine-like fashion, likely being involved in the progression of insulin resistance and type 2 diabetes. The proteomic analysis of ADEVs’ contents served to characterize the molecular fingerprint, providing potential biomarkers for diagnosis and personalized treatments for obesity-related diseases. ADEVs, adipose-derived extracellular vesicles; NG, normoglycemia; IFG, impaired fasting glucose; T2D, type 2 Diabetes [68,73,75,76]. Created with BioRender.com (accessed on 25 May 2022).

To tackle some of the concerns described above, Zhang et al. [75], using bioinformatics, recently analyzed the proteome of ADEVs; they identified adipokines that had not been previously reported further exploring those linked to obesity-related diabetes and exclusive to exosomes. Subsequently, they used these new proteins (STEAP3, NPM3, and DAD1) for in vitro and in vivo validation; they analyzed these proteins during the differentiation process of ADSCs to pre-adipocytes. Later, using a mouse genetic model of obesity and samples from subcutaneous and visceral explants, the authors found that the concentrations of NPM3 and DAD1 were higher in the wildtype control group for serum, explants, and exosomes, while no difference was found in STEAP3 [75]. These results allowed them to characterize new adipokines involved in the development of obesity and IR, and more importantly, we believe that EVs containing these proteins could be used as vehicles to promote metabolic homeostasis.

To further explore their findings on a proteomic analysis of EVs from adipocytes [66], Camino et al. [76] used explants of subcutaneous and visceral AT of subjects who were morbidly obesity (mean BMI 47.9 Kg/m^2^) versus a control group (mean BMI 26.0 Kg/m^2^). They placed the explants in a medium and collected EVs from the medium, further characterizing them into two subgroups (obese: subcutaneous and visceral; lean: subcutaneous and visceral). A different molecular footprint of visceral and subcutaneous AT vesicles was identified through proteomic-bioinformatics analyses with a stricter fold change (2 arbitrary units). Regarding the obesity group, they found a higher abundance of proteins involved in vesicular transport, catalytic, and chaperon activity in a subcutaneous depot compared to visceral AT whose enrichment is related to immunometabolic processes of obesity, such as TGFB1, CAVN1, monocyte differentiation antigen CD14, mimecan among others, all in comparison to the control group [76]. 

In addition, in [76], the authors compared plasma-derived EVs from participants with obesity and a family history of T2D vs. a control group, and found that mimecan levels were significantly increased in the obese group compared to lean subjects. In addition, the TGFB1 level was higher in obese subjects with family history of T2D. In a very interesting approach, they used Caveolin 1, which they found to be enriched in EVs derived from AT in both obese and control groups, thus using it to normalize the levels of mimecan and TGFB1, further suggesting that the EVs in circulation came from the AT. The authors concluded that TGFB1 and mimecan could be used as a potential biomarker of T2D and visceral obesity, respectively [76]. The approach used by [76] for proteomic analysis and EVs isolation method followed the recommendations of MISEV2018 [1], and should be considered for future research, as shown in Table 1.

It appears that ADEVs are essential for intercommunication both locally and distally for IR development and regulation associated with obesity; however, more studies are required to functionally analyze these adipokines and elucidate their regulatory mechanisms. In addition, some studies have found a correlation between blood EVs concentrations and their miRNA and protein content with obesity and IR [51,68,80,81,82]; briefly, to understand the molecular mechanism of insulin signaling, further information is provided below. 

## 6. PI3K/Akt/mTOR Signaling Cascade

Insulin begins its biological actions by binding it to specific receptors located in the cell membrane. After its binding, the insulin receptor undergoes conformational changes, causing its subunits to self-phosphorylate in Tyr residues. At this point, transduction pathways are activated: the 3-kinase phosphatidylinositol pathway (PI3K) and the path of mitogen-activated kinases (MAP kinases) [83]. Both pathways regulate most insulin actions associated with regulating energy metabolism, genetic expression and myogenic effects. Regarding energy metabolism, PI3K phosphorylates to PIP2 (PI3,4-bisphosphate) generating PIP3 (PI3,4,5-triphosphate). The latter then serves as a binding site for Ser kinases such as PDK1 (phosphoinositide-dependent kinase-1), and Akt or protein kinase B (PKB) [39,64]. At this point, the signaling by Akt kinase is amplified by phosphorylating several proteins involved in anabolic processes, such as mammalian target of rapamycin complex 1 (mTORC1) [83,84].

In this way, insulin signaling pathway is key to metabolic homeostasis, the target tissues of insulin are AT, skeletal muscle, liver, pancreas, and brain [85]. Finally, a key nutrient state regulator that has already been well described is mTOR [84,85]. Hence, when the PI3K/Akt/mTOR pathway is altered, in case of over expression, obesity may be developed; if obesity persists as a chronic condition, the excess of free fatty acids in blood due to increased lipolysis, can cause alteration in other non-AT tissues, for example resulting in ectopic lipids in skeletal muscle, decreasing glucose transport and glycogen synthesis, leading to imbalance in glucose metabolism [39,83,86]. 

mTOR’s role is suggested to be important in the biology and function of AT, either directly or indirectly activating adipogenesis and lipogenesis, while inhibiting lipolysis [87,88]. In addition, this pathway in turn is regulated uphill by PI3K in response to insulin or independently against stimuli of amino acids (e.g., BCAAs) [87,89]. Briefly, through the phosphorylation of downstream substrates by kinase mTORC1, the protein p70 ribosomal S6 kinase (S6K1) is activated, resulting as a negative feedback loop on PI3K/Akt which occurs due to phosphorylation on serine residues of insulin receptor substrate (IRS-1) [90].

One of the known adapter proteins in this process is the growth factor receptor-bound protein 10 (Grb 10) [91]. In this respect, Edick et al. [92] observed that knockdown of Grb 10 enhanced insulin signaling and glucose uptake in myotubes [92], suggesting this protein adaptor could modulate mTOR action in skeletal muscle and the glucose uptake in subjects with some degree of IR.

In relation to the mTORC1 function over the previous PI3K/Akt signaling, it has been reported elsewhere [85,86,87,88,89] that, due to the activation of anabolic processes, important catabolic pathways, such as autophagy, are inhibited; the results in [93,94] suggest that mTOR can be associated with IR in animal model. However, regulatory mechanisms of insulin signaling by downstream effectors of mTOR have not been entirely elucidated. To us, finding that miRNAs can negatively regulate mTOR signaling proteins while mTOR itself promotes protein synthesis of key proteins involved in metabolic pathways is interesting and opens new research opportunities. 

As mentioned previously, in the literature there is an extensive study of miRNAs contained in ADEVs [42,49,50,51,52,53,54,55,56,57,58,59,67,78,79], or different sources involved in the regulation of glucose and lipid metabolism, with an emphasis on insulin signaling [95,96]. Recently, Li et al. [97] observed that as a result of lipotoxicity derived from IR of AT, the pancreas secreted miR-29 in EVs produced in vitro after adding free fatty acids into the medium; this miRNA’s target gene is p85α, and insulin resistance might develop when p85α reduction leads to the inhibition of subunit p110 [98]. Here [97], the authors observed a reduction in p85α accompanied by a decrease in subunit p110, which is consistent with previous research showing that miR-29s inhibits PI3K, IRS-1 and Akt as part of phosphorylation mediated signaling pathways. 

A recent study by Zhao et al. (2020) showed that the liver communicates with AT through EVs, affecting adipogenesis and lipogenesis processes as a result of lipotoxicity [99], with results similar to those presented in [97]. At this point, it is of paramount importance to evaluate the mechanisms that derive from lipid overload as a direct result of IR of AT and in the onset of systemic IR in early obesity; additionally, a review database of miRNAs present in blood circulation should be constructed.

When studying adipogenesis and lipolysis, Jordan et al. [100] found a cluster miRNA-143/145 associated with these processes, as well as obesity and IR. They found miRNA-143 is a positive regulator associated to adipogenesis, which they tested using a diet-induced obesity mouse model, where miRNA-143 gene expression was inactivated and no IR was observed; additionally, over expression of miRNA-143 resulted in IR [100]. The role of miRNA 145 in obesity is less clear, although it has been associated with increased lipolysis [101]. It has been suggested that IR develops due to the production of pro-inflammatory cytokines that impact on the insulin signaling cascade [101]. However, more studies are required to elucidate the role of the cluster miRNA-143/145 from ADEVs. In circulation, miRNAs have been associated with lipid metabolism processes and glucose homeostasis in vitro or in vivo models, affecting proteins of PI3K/Akt/mTOR signaling cascade [102,103,104,105,106]; although some evidence is available, it remains unclear if miRNA expression contained in ADEVs should be analyzed independently or as a cluster to dilucidated their function/role. It is important to highlight that future studies should analyze both the genetic content and bioactive proteins, to more accurately understand their role in the regulation of metabolism [25]. Other authors also highlight the need to standardize procedures for the study of ADEVs to avoid misinterpretations [107].

## 7. Conclusions and Future Perspectives

This review recapitulated advances in the functionality research of EVs, particularly those derived from intercommunication with obese-adipose tissue. Mainly on the regulation of gene expression and signaling pathways affecting insulin sensitivity locally and systemically. Differences in results can be due to different methodological approaches; thus, the suggested mechanisms of metabolic regulation are not conclusive. In this sense, more studies are needed to analyze the content of EVs, especially bioactive proteins and miRNAs, and their impact on the development of IR in insulin-dependent tissues. Because obesity is a heterogeneous condition, the analysis and delimitation of conditions during the onset of obesity is critical. ADEVs content can be potential markers in IR, T2D, and obesity related conditions, especially when identifying activated signaling pathways regulated by phosphorylation during intercellular communication.

## Figures and Tables

**Table 1 cells-11-01831-t001:** Adipose tissue-derived extracellular vesicles and their metabolic implications.

Source *	Content	Role	Refs
AT	RBP4	Activate bone marrow macrophages	[40]
Adipocyte	mRNA (Adiponectin, resistin, PPARγ2)	Adipogenesis	[46]
Adipocyte	GPI	Lipogenesis	[49]
Adipocyte	miR-155	M1 macrophage polarization; AT IR	[52]
AT	miR-27a	Activate macrophage; IR, AT IR	[53]
Adipocyte	miR-34a	M1 macrophage polarization; adipose inflammation	[54]
ATM	miR-155	Insulin sensitivity	[55]
Plasma	miR-34a, miR-122, miR-192	Glucose and lipid metabolism	[56]
AT	miR-141-3p	Hepatic IR	[57]
Adipocyte	miR-27a	Skeletal muscle IR	[59]
Adipocyte	MALAT1	Increase food intake	[60]
AT	ABCA1, ABCG1	Macrophage foam cell generation and M1 polarization	[61]
Plasma	miR-29a	ATP production cardiomyopathy	[62]
AT	IL-6, MCP-1, MIF, RBP-4, Resistin	Hepatic IR	[63]
Adipocyte	SHH	M1 macrophage polarization; AT IR	[64]
Adipocyte	PTEN	AT IR	[65]
Adipocyte	Mimecan, Perilipin A, FABP4, TGB1, Cystatin C	AT Inflammation & IR; Biomarker	[66]
ADSCs	Arginase-1, Tyrosine Hidroxylase	M2 macrophage polarization; AT insulin sensitivity	[67]
Plasma	Perilipin A	DIO mice/Obese Humans Biomarker	[68]
Adipocyte	Adiponectin, FABP4	Adipogenesis Biomarker	[69]
Adipocyte	Specific cholesterol enrichment (sEVs)/fosfatidil serine (lEVs)	Small & Large EVs Biomarker	[70]
Adipocyte-Endothelial	Cav-1	Nutrient state Biomarker	[71]
Adipocyte	Cav-1, LPL, AQP7	Obesity Biomarker	[73]
Adipocyte	AT secretome	Exosomal adipokines footprint	[74]
AT	NPM3, STEAP3, DAD1	Obesity & Adipogenesis Biomarker	[75]
ADSCs			
Serum			
AT	TGFB1, Cav-1, FABP4, Mimecan	Obesity Biomarker	[76]
Plasma			
Adipocyte	IL-6, TNFα, MCP-1	AT & Systemic IR	[77]
AT			
ATM	miR-29a	IR (AT, myocyte and hepatocyte)	[78]
Adipocyte	miR-125a-5p, miR-296-3p, miR-298-5p, miR-351-5p	Insulin secretion & β cell function	[79]

* AT, adipose tissue; ADSCs, adipose derived stem cells; ATM, adipose tissue resident macrophages; miR, microRNA; mRNA, messenger RNA; IR, insulin resistance; DIO, diet induced obesity.

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
