# Peer review of "Extracellular Vesicles from Adipose Tissue Could Promote Metabolic Adaptation through PI3K/Akt/mTOR"

_cells, 2022, doi:10.3390/cells11111831_

Round 1

Reviewer 1 Report

General comments

EVs are considered potential  key mediators in the development of obesity. Additionally, EVs are associated with the onset and progression of obesity-related metabolic diseases. The paper describes the possible communication of EVs derived from adipose tissue with metabolically active organs, like liver or skeletal muscle, especially in terms of insulin resistance. Also the intracellular response in the insulin signalling pathway is discussed.

The idea for the article is not innovative, as there are papers discussing similar issues, but it can be considered as another voice in the discussion. The topic is described in details, the paper creates a logical whole and contains correct citations. However I think, the language could be improved.  

Specific comments:

The fragments below are cited from the text and needs to be corrected (some proposals in brackets). In some cases the structure of the sentence is incorrect and therefore it loses its sense.

Line 47 As a result,  reposition of the phosphatidyl serine to the outer side of the PM. (the verb is missing)

Line 105 On the other hand, mature adipocytes are cells that range from 50 to 100 m in diameter, those present en WAT have a single lipid (m means meters, en?)

Line 110 Interestingly, it has been reported in murine model, that exosomes from AT promote and modulate adipogenesis in adipose mesenchymal cells. Which resulted in a coordinated process in the microenvironment of cells conforming AT. (without dot between sentences?)

Line 115 During obesity, it reaches the maximum capacity of expansion of AT. At this point is developed inflammation, which now ATM produce pro-inflammatory cytokines that affect insulin signaling. Causing lipotoxicity and leading to ectopic fat accumulation in tissues such as liver and skeletal muscle, among others. (style)

Line 133 It was observed that to be promoted in skeletal muscle cells in a co-culture model.

Line 143 and affect de morphology of EVs (the)

Line 191 In which the ATM population reaches 50% of SVF [31], at this point most ATMs change from an M2 anti-inflammatory phenotype to a pro-inflammatory M1 phenotype.

Line 228 However, in relation to the possible role of miRNAs contained in ADEVs in endocrine communication with other cells in IR situation.

Line 236 They concluded that the exosomes derived from AT, act as new adipokines [42]. This section may be divided by subheadings. It should provide a concise and precise description of the experimental results, their interpretation, as well as the experimental conclusions that can be drawn.

(I guess the second and the third sentence are here by an accident)

Line  276 They validated that the expression of this miRNA was increased during adipogenesis, by bioinformatic and in vitro analyses of 3T3-L1 cells, they showed that the affected gene of this miRNA turned out to be the coactivator 1α of PPARɤ (PGC1-α),) involved in fat oxidation and mitochondrial biogenesis, in addition, TGF-β activated the secretion of miRNA-130b in adipocytes and validated in the culture of C2C12 myotubes. (the sentence is too long)

357 In another study, Eguchi et al. observed in the cultivation of adipocytes treated with EV from an obese mouse model, which promoted the activation of macrophages M1 and resulted in higher concentration of pro-inflammatory cytokines.

515 Mainly on the regulation of signaling and gene expression pathways affecting insulin sensitivity locally as a systemic.

Author Response

Response to Reviewer 1

General comments

The idea for the article is not innovative, as there are papers discussing similar issues, but it can be considered as another voice in the discussion. The topic is described in details, the paper creates a logical whole and contains correct citations. However I think, the language could be improved.

Dear Reviewer, thank you for your comments. We certainly improved the discussion throughout the manuscript and corrected the language.

Specific comments:

Point 1: Line 47 As a result,  reposition of the phosphatidyl serine to the outer side of the PM. (the verb is missing)

Response 1:  Thank you for your suggestion, according to “Line 51 “As a result, phosphatidyl serine is repositioned to the outer side of the PM

Point 2: Line 105 On the other hand, mature adipocytes are cells that range from 50 to 100 m in diameter, those present en WAT have a single lipid (m means meters, en?)

Response 2:  Thank you for detect this missing, “Line 107” Mature adipocytes are cells that range from 50 to 100 μm in diameter

Point 3: Line 110 Interestingly, it has been reported in murine model, that exosomes from AT promote and modulate adipogenesis in adipose mesenchymal cells. Which resulted in a coordinated process in the microenvironment of cells conforming AT. (without dot between sentences?)

Response 3:  We appreciate your suggestion, according to “Line 112”  Interestingly, using a murine model Zhang et.al., [36] and Son et.al., [37], showed that exosomes from AT promote and modulate adipogenesis in adipose mesenchymal cells, resulting in a coordinated process inside the microenvironment of cells conforming AT.

Point 4: Line 115 During obesity, it reaches the maximum capacity of expansion of AT. At this point is developed inflammation, which now ATM produce pro-inflammatory cytokines that affect insulin signaling. Causing lipotoxicity and leading to ectopic fat accumulation in tissues such as liver and skeletal muscle, among others. (style)

Response 4: Corrections has been made, thank youLine 117During obesity the maximum expansion capacity of the adipocyte is reached, now ATM produces pro-inflammatory cytokines that affect insulin signaling, causing lipotoxicity and leading to ectopic fat accumulation in tissues like liver and skeletal muscle.

Point 5: Line 133 It was observed that to be promoted in skeletal muscle cells in a co-culture model.

Response 5: Once again, we appreciate for your suggestion for that incomplete sentence, we have changed for “Line 137” Dietze et. al., [43] studied in vitro adipocyte communication from breast tissue explants of women with BMI 22-29Kg/m2 alongside myocytes from normal weight individuals, and they found that when pro-inflammatory cytokines and resistin were placed on the medium, no difference in skeletal muscle IR development was observed.    

Point 6: Line 143 and affect de morphology of EVs (the)

Response 6:  Thank you for your suggestion, “Line 147” Those conditions simulated an obesity environment affecting the morphology of EVs.

Point 7: Line 191 In which the ATM population reaches 50% of SVF [31], at this point most ATMs change from an M2 anti-inflammatory phenotype to a pro-inflammatory M1 phenotype.

Response 7:  We have restructured the sentence “Line 195” When the ATM population reaches 50% of SVF [31], most ATM change from an anti-inflammatory M2 phenotype to a pro-inflammatory M1 phenotype, leading to high concentrations of cytokines and EVs containing miRNAs being secreted, which in the end can trigger IR.

Point 8:  Line 228 However, in relation to the possible role of miRNAs contained in ADEVs in endocrine communication with other cells in IR situation.

Response 8:  Thank you, we improved the wording “Line 231” Here we explore the role of miRNAs contained in ADEVs in IR, through endocrine-like communication with other cells.

Point 9:  Line 236 They concluded that the exosomes derived from AT, act as new adipokines [42]. This section may be divided by subheadings. It should provide a concise and precise description of the experimental results, their interpretation, as well as the experimental conclusions that can be drawn.

(I guess the second and the third sentence are here by an accident)

Response 9:  Thank you for detecting the mistake, we changed to “Line 237” Authors concluded that the exosomes derived from AT, act as new adipokines [42]. (without second and the third sentence)

Point 10:  Line  276 They validated that the expression of this miRNA was increased during adipogenesis, by bioinformatic and in vitro analyses of 3T3-L1 cells, they showed that the affected gene of this miRNA turned out to be the coactivator 1α of PPARɤ (PGC1-α),) involved in fat oxidation and mitochondrial biogenesis, in addition, TGF-β activated the secretion of miRNA-130b in adipocytes and validated in the culture of C2C12 myotubes. (the sentence is too long)

Response 10:  In deed, we have changed the sentence. Thank you for suggesting. “Line 269”  They validated that the expression of this miRNA was increased during adipogenesis, by bioinformatic and in vitro analyses of 3T3-L1 cells. They showed that the target gene of miRNA130-b was the coactivator 1α of PPARɤ (PGC1-α), involved in fatty acid oxidation and mitochondrial biogenesis.

Point 11:  Line 357 In another study, Eguchi et al. observed in the cultivation of adipocytes treated with EV from an obese mouse model, which promoted the activation of macrophages M1 and resulted in higher concentration of pro-inflammatory cytokines.

Response 11:    Thank you for suggesting changes about style, It was as follows “Line 351” In another study, Eguchi et.al., [68] obtained EVs from plasma of an obese mouse model and treated a culture of adipocytes with these EVs promoting the activation of macrophages M1, which resulted in higher concentration of pro-inflammatory cytokines.

Point 12:  Line 515 Mainly on the regulation of signaling and gene expression pathways affecting insulin sensitivity locally as a systemic.

Response 11:     Once again, thank you for your remark. We have changed to “Line 535” Mainly on the regulation of gene expression and signaling pathways affecting insulin sensitivity locally and systemically.

Reviewer 2 Report

The manuscript provides some information regarding the biogenesis and function of EVs, with a focus on adipose-associated EVs. However, a large part of the manuscript is irrelevant to the topic, but is to summarize basic knowledges in other aspects, such as the adipose biology, insulin signalling, which is not supposed to be the focus of the review. Furthermore, the review is not comprehensive, that although the authors mentioned that EVs are classified to exosomes and MVs, depending on their size, in the following discussion, only studies on exosome are included. The manuscript also lacks logic to me. It is unknown whether the narration is organized according to different types of the source cells of the EVs, or according to the destination cells, or by the contents of the EVs? And it is also unclear why “ PI3K/Akt/mTOR signalling cascade” is enlisted as a separate section at the end of the review.

Major points:

  1. references are missing in a number of sentences. That is to say reference(s) shall be provided immediately at the end of the sentence once a statement is made.

For example: (1) line 46 on page 2: “The changes are induced by flippases and translocases enzymes that are activated by the increase in the cytosolic concentration of Ca+2”. (2)  line 55 on page 2: “The biological signal is transmitted upon EVs binding to specific ligands, thus activating signaling cascades or merging directly with the PM in order to enter into cytoplasm.” (3) line 74 p2 “Then, regarding energy homeostasis, the tissues mostly involved are the brain, pancreas, liver, skeletal muscle and adipose tissue”.

  1. Line 133 page 3 “It was observed that to be promoted in skeletal muscle cells in a co-culture model”. What does this sentence mean? There are a large number of sentences with the same problem that the language needs extensive review and editing.
  2. The last paragraph on page 3, again almost no reference is provided in this whole paragraph. The only one provided is a retracted one (ref 44)!
  3. The review is not comprehensive, that although the authors mentioned that EVs are classified to exosomes and MVs, depending on their size, in the following discussion, only studies on exosome are included. Furthermore, as a mini-review, the manuscript shall be compact. In review article, each paragraph shall talk about one major point by including several relevant studies. However, in this manuscript, in each one paragraph, very often only one piece of work is described. This is not what a review article shall be like.
  4. The manuscript also lacks logic. It is unknown whether the narration is organized according to different types of the source cells of the EVs, or according to the destination cells, or by the contents of the EVs? It is also unclear why “ PI3K/Akt/mTOR signalling cascade” is enlisted as a separate section at the end of the review?

Author Response

General comments

The manuscript provides some information regarding the biogenesis and function of EVs, with a focus on adipose-associated EVs. However, a large part of the manuscript is irrelevant to the topic, but is to summarize basic knowledges in other aspects, such as the adipose biology, insulin signalling, which is not supposed to be the focus of the review. Furthermore, the review is not comprehensive, that although the authors mentioned that EVs are classified to exosomes and MVs, depending on their size, in the following discussion, only studies on exosome are included. The manuscript also lacks logic to me. It is unknown whether the narration is organized according to different types of the source cells of the EVs, or according to the destination cells, or by the contents of the EVs? And it is also unclear why “ PI3K/Akt/mTOR signalling cascade” is enlisted as a separate section at the end of the review.

Thanks for your observations, the authors believe that it is important to comment on the relevant aspects of the biology of adipose tissue and the generation of ADEVs, so that it is understood how communication between adipose tissue and muscle can be established. It is also relevant to understand the possible signal transduction mechanism involved (PI3K/Akt/mTOR). It is important to highlight the role of the signaling pathway (PI3K/Akt/mTOR) since it is one of the candidates, and in particular for this review the candidate proposed to exert the signal in muscle tissue (Insulin resistance). It is for the above that the order of the paper begins with the biogenesis of the EV's and key concepts of adipose tissue, to continue with the formation of the ADEV's as a communication vehicle and finally the phosphorylation route that the evidence seems to point to. as the candidate to perform signal transduction to muscle.

Specific comments:

Point 1: references are missing in a number of sentences. That is to say reference(s) shall be provided immediately at the end of the sentence once a statement is made

Response 1: Thank you for your comments on the manuscript. We have indeed rearranged the citation by sentence/idea throughout the document and we think it has improved.

  (1) line 46 on page 2: “The changes are induced by flippases and translocases enzymes that are activated by the increase in the cytosolic concentration of Ca+2”.

Response (1):  “Line 49” The changes are induced by flippases and translocases enzymes that are activated by the increase in the cytosolic concentration of Ca+2 [11]. As a result, phosphatidyl serine is re-positioned to the outer side of the PM. At the same time, the increase in calcium activates Rho-II kinase proteins (responsible for phosphorylating myosin chains) [12], followed by binding with actin. These proteins combined with enzymes such as syntenin-1 and syn-decane-1, cause reorganization and contraction of the cytoskeleton, protrusion of PM, and finally detachment of these vesicles into the extracellular medium [13].

(2)  line 55 on page 2: “The biological signal is transmitted upon EVs binding to specific ligands, thus activating signaling cascades or merging directly with the PM in order to enter into cytoplasm.”

Response (2):  “Line 57” The biological signal is transmitted upon EVs binding to specific ligands, activating signaling cascades or merging directly with the PM to enter the cytoplasm [11]. Another way to uptake EVs, is through endocytosis or phagocytosis, with a lipid membrane or raft receptor, generating multivesicular bodies that eventually release their functional content to the cytoplasm or are transported to the nucleus [14-16].

(3) line 74 p2 “Then, regarding energy homeostasis, the tissues mostly involved are the brain, pancreas, liver, skeletal muscle and adipose tissue”.

Response (3):  “Line 76” . In particular, regulation of energy homeostasis takes place mainly in the brain, pancreas, liver, skeletal muscle, and adipose tissue [23].

Point 2:  Line 133 page 3 “It was observed that to be promoted in skeletal muscle cells in a co-culture model”. What does this sentence mean? There are a large number of sentences with the same problem that the language needs extensive review and editing

Response 2:  Thank you for the suggestion, indeed changes were made throughout the manuscript regarding the wording and coherence of the ideas.(e.g. “Line 137” Dietze et. al., [43] studied in vitro adipocyte communication from breast tissue explants of women with BMI 22-29Kg/m2 alongside myocytes from normal weight individuals, and they found that when pro-inflammatory cytokines and resistin were placed on the medium, no difference in skeletal muscle IR development was observed.  This suggested that the onset and progression of obesity related IR is not only influenced by hormones but by other adipocyte-typical factors released into the medium. This opened a new research area to identify the "other factors", which we now know as adipokines.

Point 3: The last paragraph on page 3, again almost no reference is provided in this whole paragraph. The only one provided is a retracted one (ref 44)!

Response 3:  We appreciate your suggestion, according to “Line 144 We discuss the results of the above-mentioned reference, in order to present the general aspects that allowed us to further investigate adipocyte-derived vesicles” as follows; Later, Aoki et. al., [44] analyzed EVs secreted by adipocytes in vitro. They found that larger adipocytes supplemented with high concentrations of insulin and glucose in the medium, secreted both larger size and amounts of EVs, which the authors called adiposomes. Those conditions simulated an obesity environment affecting the morphology of EVs. However, other factors that also interfered were the positive and negative controls used in the experiment (N-acetyl cysteine and TNF-α, respectively), showing that various intracellular conditions can trigger a functional response mediated by EVs [44].

Point 4: The review is not comprehensive, that although the authors mentioned that EVs are classified to exosomes and MVs, depending on their size, in the following discussion, only studies on exosome are included. Furthermore, as a mini-review, the manuscript shall be compact. In review article, each paragraph shall talk about one major point by including several relevant studies. However, in this manuscript, in each one paragraph, very often only one piece of work is described. This is not what a review article shall be like.

Response 4: A review of the literature was carried out and although the classification of extracellular vesicles is still under discussion, some consensus is being formed depending on their size or their origins. Under this optics they are mentioned as EVs throughout the manuscript. Respecting the names originally attributed to them by the authors of the aforementioned works on paper, most of the available literature names them as exosomes based on their size nomenclature. We have worked on the document and we believe that in response to the comments of the reviewers it has improved substantially, including a change in the proposed title.

Point 5: The manuscript also lacks logic. It is unknown whether the narration is organized according to different types of the source cells of the EVs, or according to the destination cells, or by the contents of the EVs? It is also unclear why “ PI3K/Akt/mTOR signalling cascade” is enlisted as a separate section at the end of the review?

Response 5: Thank you very much for your comments. A table was included in the document to identify the origin, content and function of the ADEVs in order to better discuss and organize the information presented. As shown in Table 1; “Line 130”. Regarding the organization of the document, we believe this is clarified in the general comments and throughout the new version of the improved document.

Reviewer 3 Report

In this review, extracellular vesicles (EVs) are described as potential key mediators in the physiopathology of obesity. EVs derived from adipose tissue contribute to the development of a local inflammatory response and furthermore EVs are associated with the onset and progression of metabolic diseases such as insulin resistance. In this review a possible crosstalk between EVs released from adipose tissue with metabolically active organs and the intracellular response in the insulin signaling pathway is described.

This review describes the influence of extracellular vesicles on metabolic diseases such as insulin resistance, which is a very challenging topic.

Major points

However, in this review several aspects of this topic are put together, whereby the interrelations were not clarified.

Authors describe effects of several miRNAs or proteins of ADEVs (EVs derived from adipose tissue), which induce e.g. maturation of preadipocytes or insulin signaling.

However, adipocyte-derived exosomes may participate in the recruitment and M1 polarization of macrophages, but also in the induction of insulin resistance. Furthermore, exosomes released from macrophages are involved in obesity-related diseases. In this review, all these points are mixed together and no clear structure is identifiable.

The direct source of EVs and their targets remain sometimes unclear.

A table including the source, the content, and functions of different kinds EVs would be helpful.

The legend of Figure 1 is too short. For example, the symbols of the phenotypes are not explained.

The influence of EVs on insulin signaling, which is mainly involved in the development of IR, is described only very briefly.

It remains unclear if analysis of exosomes can be used as a diagnostic method for the early diagnosis of IR or as target for the treatment of the disease.

Minor points

List of abbreviations should be included into the manuscript to make the reading of the article easier.

English writing and grammar have to be revised. Some sentences are not understandable (e.g. line 133).

Author Response

General comments

This review describes the influence of extracellular vesicles on metabolic diseases such as insulin resistance, which is a very challenging topic.

Dear Reviewer, thank you for your comments. Indeed, research on extracellular vesicles is practically recent. Therefore, research on the functions of these vesicles is difficult to homologate. In this respect, the aim of our review was to provide an overview of the origin and metabolic implications of adipose tissue-derived vesicles. Due to the high prevalence of obesity worldwide, we present some of the effects that adipose-derived extracellular vesicles may have on the development or diagnosis of obesity-associated diseases, such as insulin resistance.

Major points

Point 1:

However, in this review several aspects of this topic are put together, whereby the interrelations were not clarified. Authors describe effects of several miRNAs or proteins of ADEVs (EVs derived from adipose tissue), which induce e.g. maturation of preadipocytes or insulin signaling.

However, adipocyte-derived exosomes may participate in the recruitment and M1 polarization of macrophages, but also in the induction of insulin resistance. Furthermore, exosomes released from macrophages are involved in obesity-related diseases. In this review, all these points are mixed together and no clear structure is identifiable.

Response 1: Thank you for the comments, indeed changes were made throughout the manuscript regarding the wording and coherence of the ideas. Additionally, the results of different studies about the cell types that comprise the adipose tissue were also included to demonstrate the possible local and systemic regulation in which extracellular vesicles are involved. Particularly, Adipose resident macrophages (“Line “195”; “Line 206”; “Line 212”; “Line 224”; “Line 321”) In addition, an attempt was made to separate the information, to summarize it in the form of a figure (“Line 393”) and to discuss the general aspects of each study in question with the aim of suggesting the role of metabolic agents that the vesicles derived from adipose tissue in obesity may exert on the development of insulin resistance.

Point 2: 

The direct source of EVs and their targets remain sometimes unclear. A table including the source, the content, and functions of different kinds EVs would be helpful.

Response 2: Thank you for your valuable suggestions. In this respect a table was included in the document to identify the origin, content and function of the ADEVs in order to better discuss and organize the information presented. As shown in Table 1; “Line 130”.

Point 3:

The legend of Figure 1 is too short. For example, the symbols of the phenotypes are not explained.

Response 3: Once again, thank you for your suggestions to the manuscript. As requested, as in the previous section, we have modified the Figure 1 (“Line 393”). In which we added the phenotypes and complete description of the approach that we intend to present in the legend of the same figure.

Point 4:

The influence of EVs on insulin signaling, which is mainly involved in the development of IR, is described only very briefly.

Response 4: Thanks for the feedback. In fact, most of the sentences were reinterpreted to emphasize the possible regulation of the insulin signaling pathway by EVs. (This focuses the reader on miRNAs/proteins that may regulate the development of insulin resistance. Mainly those molecules that are contained in EVs derived from adipose tissue.

“Line 244”, They also found that miRNA-155 negatively regulated the expression of PPARɤ and, this in turn the protein GLUT4, inferring that the effects had an impact on the cascade of insulin signaling in adipocytes, myotubes and hepatocytes).

“Line 257”, exosomes were captured by in vitro hepatocytes and miRNA expression was accompanied by increased phosphorylation of the Akt protein.

“Line 263”, miRNA-141-3p affected PTEN protein, a phosphatase that dephosphorylates PI3K, involved in the insulin signaling cascade.

“Line 311”, These data suggest that ADEVs may secrete pro-inflammatory cytokines that negatively affect the phosphorylation of Akt in the liver for the development of IR.

Point 5:

It remains unclear if analysis of exosomes can be used as a diagnostic method for the early diagnosis of IR or as target for the treatment of the disease.

Response 5: Thank you for your comments. Indeed, the studies in which the molecular fingerprint in obesity can be determined by proteomic analysis of EVs are discussed. That allows to propose them as a potential biomarker and as a possible treatment for metabolic disorders associated with obesity. Similarly, in Figure 1 together with its legend, we can visualize precisely this proposal for the analysis of EVs as biomarkers and as a treatment that we intend to discuss starting on page 8 (“Line 333”).and Figure 1 “Line 394”

Minor points

Point 1:  List of abbreviations should be included into the manuscript to make the reading of the article easier.

Response 1: Thanks for the valuable suggestion, we include at the end of the document the list of abbreviations that supports the reader throughout the manuscript.(“Line 793”)

Point 2: English writing and grammar have to be revised. Some sentences are not understandable (e.g. line 133).

Response 2: Thank you for the suggestion, indeed changes were made throughout the manuscript regarding the wording and coherence of the ideas.(e.g. “Line 137” Dietze et. al., [43] studied in vitro adipocyte communication from breast tissue explants of women with BMI 22-29Kg/m2 alongside myocytes from normal weight individuals, and they found that when pro-inflammatory cytokines and resistin were placed on the medium, no difference in skeletal muscle IR development was observed. This suggested that the onset and progression of obesity related IR are not only influenced by hormones but by other adipocyte-typical factors released into the medium. This opened a new research area to identify the "other factors", which we now know as adipokines.

Reviewer 4 Report

The manuscript addresses a very interesting topic and presents important data from the literature. This review presents the effects of extracellular vesicles on metabolism.

However, some points should be better discussed:

1) I suggest the authors to remove the term "biomolecules" from the title, as they address only the effects of extracellular vesicles;

2) The manuscript needs an English review; it has some mistakes and some sentences appear incomplete

e.g.:

Sintenin x syntenin

Line 105 “On the other hand, mature adipocytes are cells that range from 50 to 100 m in diameter”

Line 228 “However, in relation to the possible role of miRNAs contained in ADEVs in endocrine communication with other cells in IR situation.”

3) The authors must review concepts about extracellular vesicles. Currently, vesicles are defined as small or large (according to the size).

4) When authors cited the effects of extracellular vesicles on macrophages, it may be interesting to cite their effects on monocytes.

5) It would be interesting if the authors discuss more about Zhang's study: Lines: 409 - 416

Author Response

General comments

The manuscript addresses a very interesting topic and presents important data from the literature. This review presents the effects of extracellular vesicles on metabolism. However, some points should be better discussed:

Point 1:

I suggest the authors to remove the term "biomolecules" from the title, as they address only the effects of extracellular vesicles;

Response 1: Dear colleague, thank you for your comment and suggestion. In fact the manuscript focuses on the study of adipose tissue-derived EVs in the regulation of insulin resistance and their potential use as biomarkers of obesity. We have changed the title to "Extracellular vesicles from adipose tissue could promote metabolic adaptation through PI3K/Akt/mTOR”

Point 2:

The manuscript needs an English review; it has some mistakes and some sentences appear incomplete

Response 2: Thank you for the comments, indeed changes were made throughout the manuscript regarding the wording and coherence of the ideas. We think it has improved.

e.g.: Sintenin x Syntenin;  Changed as follows “Line 54” syntenin-1 and syndecane-1

Line 105 “On the other hand, mature adipocytes are cells that range from 50 to 100 m in diameter”

Line 107”, Mature adipocytes are cells that range from 50 to 100 μm in diameter.

Line 228 “However, in relation to the possible role of miRNAs contained in ADEVs in endocrine communication with other cells in IR situation.”

“Line 231” Here we explore the role of miRNAs contained in ADEVs in IR, through endocrine-like communication with other cells. Particularly, Thomou et. al., [42] showed that inactivation of ADipo, decreased the expression of circulating miRNAs; additionally, by administering BAT exosomes to an in vivo model restored the metabolic alterations associated with IR (vs. control group).

Point 3: The authors must review concepts about extracellular vesicles. Currently, vesicles are defined as small or large (according to the size).

Response 3:. Thank you for your suggestion, we have changed to “Line 26” Traditionally, EVs are classified according to their biogenesis in two subgroups, exosomes and ectosomes; the former are derived from endosomal pathway and have a size range of 40 to 160 nm. In contrast, the latter are derived from the plasma membrane and typically measure 50-1000 nm [1-3]; depending on their size, they can also be called small or large EVs, respectively [4].

Point 4:  When authors cited the effects of extracellular vesicles on macrophages, it may be interesting to cite their effects on monocytes.

Response 4: We appreciate your suggestions and comments. Indeed we agree that regulation of insulin resistance by cytokines, or other subpopulations of immune system cells in obesity is relevant. However, given the focus presented regarding adipose tissue-derived EVs on the PI3K/Akt/mTOR signaling pathway, we do not include that topic, which is more than interesting.

Point 5:  It would be interesting if the authors discuss more about Zhang's study: Lines: 409 – 416

Response 5: Thank you for your comments and suggestion about Zhang´s study. We improved our discussion and highlight their study findings at legend´s Figure 1 as potential treatment of obesity from ADEVs.

We have changed “Line 419-428” Zhang et.al., [75] using bioinformatics recently analyzed the proteome of ADEVs; they identified adipokines that had not been previously reported further exploring those linked to obesity-related diabetes and exclusive to exosomes. Subsequently, they used these new proteins (STEAP3, NPM3, and DAD1) for in vitro and in vivo validation; they analyzed these proteins during the differentiation process of ADSCs to pre-adipocytes. Later, using a mice genetic model of obesity and samples from subcutaneous and visceral explants, authors found that the concentration of NPM3 and DAD1 was higher in the wildtype control group for serum, explants, and exosomes, no difference was found in STEAP3 [75]. These results allowed them to characterize new adipokines involved in the development of obesity and IR, and more importantly, we believe that EVs containing these proteins could be used as vehicles to promote metabolic homeostasis.

Round 2

Reviewer 2 Report

The authors have in this revised version appropriately addressed most of the concerns of mine. 

Reviewer 3 Report

accept in present form